# Prior to Implementation of Digital Pathology—Assessment of Expectations among Staff by Means of Normalization Process Theory

**DOI:** 10.3390/ijerph19127253

**Published:** 2022-06-14

**Authors:** Minne L. N. Mikkelsen, Marianne H. Frederiksen, Niels Marcussen, Bethany Williams, Kristian Kidholm

**Affiliations:** 1CIMT—Centre for Innovative Medical Technology, Odense University Hospital, 5000 Odense, Denmark; kristian.kidholm@rsyd.dk; 2Department of Pathology, Hospital Sønderjylland, University Hospital of Southern Denmark, 6200 Aabenraa, Denmark; niels.marcussen@rsyd.dk; 3Department of Pathology, Odense University Hospital, 5000 Odense, Denmark; 4Department of Business and Management, University of Southern Denmark, 5230 Odense, Denmark; mha@sam.sdu.dk; 5Department of Histopathology, Leeds Teaching Hospitals NHS Trust and the University of Leeds, Leeds LS2 9JT, UK; bethany.williams2@nhs.net

**Keywords:** whole slide imaging (WSI), technology, implementation, digital pathology, normalization measure development tool (NoMAD), management of digital transformation

## Abstract

The Region of Southern Denmark is the first in Denmark to implement digital pathology (DIPA), starting at the end of 2020. The DIPA process involves changes in workflow, and the pathologist will have to diagnose based on digital whole slide imaging instead of through the traditional use of the conventional light microscope and glass slides. In addition, in the laboratory, the employees will have to implement one more step to their workflow—scanning of tissue. The aim of our study was to assess the expectations and readiness among employees and management towards the implementation of DIPA, including their thoughts and motivations for starting to use DIPA. We used a mixed-method approach. Based on the findings derived from 18 semi-structured interviews with employees from the region’s departments of pathology, we designed a questionnaire, including questions from the normalization measure development tool. The questionnaires were e-mailed to 181 employees. Of these employees, 131 responded to the survey. Overall, they reported feeling sufficiently tech-savvy to be able to use DIPA, and they had high expectations as well as motivation and readiness for the upcoming changes. However, the employees were skeptical regarding the allocation of resources, and few were aware of reports about the effects of DIPA. Based on the findings, it seems to be important to provide not only a thorough introduction to the new intervention and the changes it will entail, but also to continue to ensure that the staff know how it works and why it is necessary to implement.

## 1. Introduction

Digital pathology (DIPA), or whole slide imaging (WSI), encompasses the digitization of entire histology slides. The WSI approach was first described by Wetzel et al. in 1999 [1,2]. Digitization of pathology includes four sequential parts, as follows: image acquisition (scanning), storing, editing, and displaying of images [3]. The DIPA process has led to new opportunities not possible using conventional microscopes, including digital collaboration or telepathology, integration with electronic workflows and health records, and diagnostic support based on computational tools, such as artificial intelligence (AI) [2].

Until now, pathologists at the hospitals in the Region of Southern Denmark have been receiving glass slides and viewing them with a light microscope. If a second opinion is needed and more pathologists need to view the image, supporting staff prepare the slides for shipment, and these are then transported to another pathologist, who receives the slides and makes the assessment. This involves risks of delay for the patient and damage to the glass slides during transportation [4]. The use of DIPA could streamline this process.

The implementation of DIPA is expected to form the basis of pathology image-based AI and, thereby, release more resources. This makes it possible to enhance flexibility across the pathology departments in the region and, in that way, ensure consistent response time across the region. It makes it easier to confer with colleagues at the other departments, and both easier and quicker to share cases. The Region of Southern Denmark wishes to preserve the four departments in the region—as well as the smaller ones that could be difficult to fully occupy with pathologists—and this is made possible through the enhanced flexibility across the region due to the implementation of DIPA [5,6,7]. Furthermore, it might make the workplaces more attractive for pathologists and junior doctors, as well as aid recruitment of new doctors within pathology.

The scientific discipline that studies methods and strategies for facilitating the uptake of evidence-based practice including new workflows in clinical departments into routine daily practice is called implementation science. Implementation research is a relatively new field of science and is based on empirical studies showing that it takes, on average, 17 years for evidence-based practices to be incorporated into routine general practice in healthcare—this is called the research-to-practice gap [8,9]. Problems with implementation in the healthcare system may arise at multiple levels, such as at patient level, provider team or group level, organizational level, or market/policy level [10]. Furthermore, it has been shown that many implementation studies do not use a theory [11].

On this basis, we found that the implementation of DIPA in the Region of Southern Denmark from 2020 onwards was a relevant case for an implementation study of digital transformation in the healthcare system as it was the first region starting to implement DIPA [5].

The Region of Southern Denmark includes approximately 1.228 million citizens. Geographically, the region includes the south of Jutland, Funen and the islands south of Funen, and it has five hospitals of which four have a department of pathology [12,13,14]. All four departments are involved and included in the digitalization of pathology in the region. The total number of certified pathologists is around 52, and there are about 24 residents.

The departments of pathology carry out diagnostics of diseases on the basis of examinations of organs, tissues, and cells. The work focuses on five core tasks:-Diagnosis of cancer;-Diagnosis of noncancers;-Screening;-Monitoring;-Medical autopsies [15].

To be able to solve these tasks the departmental staff consists of pathologists, laboratory technicians, molecular biologists, autopsy assistants, and secretaries.

It was politically decided in 2019 in the Region of Southern Denmark that parts of the pathology should be digitized. The digitalization encompasses histology only, and specifically excludes cytology for the present [16]. The basis for implementing DIPA in the region was the expectation that the need for pathology tests will increase in the future [5]. The Region of Southern Denmark expects an annual increase in the number of tests of 4.5% [16]. This increase is related to both the expected increase in the number of patients with cancer and the political focus on improving the treatment pathways for patients with cancer [17]. At the same time, recruitment of new pathologists is a challenge because of a lack of consultants in pathology in Denmark. As part of the call for a new regional IT solution for DIPA in the Region of Southern Denmark, three goals concerning the implementation of DIPA were appointed, as follows [5]:**Ensuring the future of pathology in the region.** This objective relates to the guarantee of rapid turnaround of patient pathology samples, even though the number of tests is increasing, and that the potential of AI can be exploited.**Effective sharing of data between the hospitals.** This objective relates to the fact that digitization can lead to easy sharing of data between the pathological departments and the hospitals, and the fact that diagnostics can be made based on digital images across the departments.**High level of reliability.** This objective relates to easy access to digital images and low transaction costs of data.

During the process of planning the implementation of DIPA that began in 2017, several user groups have been involved in connection with defining requirements, evaluating offers, and preparing technology implementation. Table 1, below, describes the steps in the implementation process based on information from the management at one of the departments of pathology.

For the decision of choosing the technical equipment, the region followed the EU public procurement rules. Especially with regards to scanners, testing was made in connection with the decision of selecting the right machines. A requirements specification was made by end-users and IT employees for both scanners and the image management system [18].

To understand the impact of the implementation and transition from analogue working methods to DIPA for the employees at the departments of pathology, it is necessary to know exactly what is going to change in the different workflows.

The use of DIPA alters steps in the workflow, and some of the human tasks will be replaced by automatic digital transmission in the distribution of glass slides from the laboratory to the pathologists. The glass slides will be put into a scanner, and the digital images will be assessed on a screen by the pathologist. In case consultation by a sub-specialist pathologist is needed, the specialist receives a digital request to attend a shared viewing session. The employees at the departments were introduced to and educated about their new roles in the DIPA workflow by superusers from each group of the professions, who had completed a one/two-day course taking place at the software suppliers. Back at the departments, the superusers would then teach their colleagues. Furthermore, a consultant from the software supplier visited the departments from time to time and offered assistance. In the laboratory, many of the laboratory technicians were introduce to DIPA and educated on its usage by peer-to-peer training.

The aim for this study was to assess expectations and readiness for the implementation of DIPA among all professional groups at the four departments of pathology in the Region of Southern Denmark prior to the implementation.

Digitalization of hospitals is a general change involving many technologies that will change the workflow in many clinical hospital departments. It is the hope that the experiences from the departments of pathology can provide valuable information about the implementation of digital technologies in hospitals in general and, thereby, be relevant for informing such implementation processes in the coming years. Furthermore, we think this study might also be important for future implementations of DIPA in all western countries. 

## 2. Materials and Methods

In this study, we used a mixed-methods approach to measure expectations and readiness among the staff at the departments of pathology in the Region of Southern Denmark prior to the implementation of DIPA.

Observations from one of the departments of pathology gave the researchers insight into the concrete impending changes. Furthermore, a demonstration of the software by the supplier, Sectra Denmark A/S, provided an overview of the new digital program that the pathologists should use.

Initial interviews were carried out with employees from each of the professional groups (pathologists, laboratory technicians, secretaries, and managers) at the departments for the purpose of illuminating what they thought could be important in relation to the impending transformation process. Moreover, to be able to design a questionnaire, it was necessary to know the themes which could be substantial to the staff.

For the semi-structured interviews two interview guides were designed, as follows: one for employees and one for the management. Both guides were designed based on McKinsey’s 7-S framework [19]. This framework includes seven internal factors that need to align for changes in the organization to be successful, as follows: strategy, structure, skills, systems, shared values, (management) style, and staff. It can be used in situations where it is expedient to examine how various parts of an organization work together [20].

The interview guides included five overall themes that were consistent with McKinsey’s 7-S framework, as follows: view of DIPA (strategy), the implementation process (style), expectations for working with DIPA (staff and skills), expectations of the everyday work life with DIPA (systems and structure) and the motivation for pathology in general (shared values). The interviewer also focused on understanding the daily workflow and tasks for each professional groups before DIPA, and the expected new workflow and tasks after the implementation of DIPA.

A questionnaire survey was developed based on the trends and topics that were identified through the semi-structured interviews and also included questions to identify characteristics of respondents, i.e.,: age, gender, profession, and seniority. In addition, the normalization process theory (NPT) tool, normalization measure development questionnaire (NoMAD) part C, was incorporated with its 20 questions into the questionnaire survey [21]. The NoMAD is a validated questionnaire for use when measuring implementation processes from the perspective of professionals involved in the implementation with interventions in healthcare [22,23]. The NPT is an internationally recognized theory of implementation [24] that focusses on the understanding of how interventions become integrated in the daily workflow through implementation. The theory focusses on four core constructs, as follows: coherence, cognitive participation, collective action, and reflexive monitoring, through which the integration is proposed to occur [22,24,25,26]. The four core constructs includes 16 sub-constructs. For an explanation of these, see Table A2.

Additionally, more specific questions about DIPA were added, such as whether the employee received training in DIPA, expectations to ergonomics after DIPA, positive and negative aspects of DIPA, and whether the respondents felt safe about working with DIPA.

### 2.1. Sampling and Recruitment

Interviews were carried out at two of the departments of pathology. The chosen departments were different both geographically and in terms of size. Both heads of department asked employees from each professional groups to participate.

For the questionnaire survey, we wanted to include all employees at the four departments of pathology whose workflow would be affected by DIPA. The management at the departments gave their approval to conduct the survey. Furthermore, they sent a list with the e-mail addresses of the employees who were to work with DIPA.

### 2.2. Data Collection

The departments had commenced DIPA practices at different dates due to differences in their readiness of equipment, software, etc. The first department to have commence was at the end of November 2020. Then, another department followed in December 2020. The last department to begin was in spring 2021. Furthermore, the departments implemented DIPA with varying speed, meaning that not all employees started using DIPA at the same date. It could take months from the first employees at a department using DIPA to the last employees were also using DIPA. This made it impossible to ensure that all respondents and interviewees had been asked exactly 3 months before their commencement. All interviews were conducted before commencement or one day after for all the interviewees. The majority of employees received and answered the questionnaire before individual commencement.

Observations were primarily carried out at one department where the “old” workflow was observed three months before commencement at that department. Furthermore, during fieldwork at the department, the new workstations were also presented.

The semi-structured interviews were all conducted by the same person, recorded by means of a Dictaphone, and afterwards transcribed by the interviewer. Most of the interviews were carried out at the interviewees’ workplace, and a few were made via video conference. In all, 18 interviews were carried out in the period from November 2020 to January 2021, and each interview lasted between 33 and 101 min.

The questionnaire was sent to participants’ working e-mail addresses. In February 2021 it was sent to three of the four departments of pathology. Later, the last department agreed to participate in the project, and, in the beginning of March 2021, the questionnaire was sent to employees at that department. After two and four weeks, a reminder was sent. The data collection was closed after five weeks and four days. This meant that for some employees and departments it was before and for others it was after commencement of DIPA. 

All participants consented to participation and were assured that their anonymity and confidentiality would be protected.

### 2.3. Data Analysis

The interview data were stored in OPEN [27]—a safe database at Odense University Hospital intended for research data, and coded using the qualitative analyzing software NVivo 11. Furthermore, quotes were coded into subcategories in an Excel sheet to enable comparison of the data across the different groups of professions. Based on the comparisons, it was decided which topics should be included in the questionnaire.

The STATA 17. software was used for the analysis of the quantitative data. Interferential statistic (*χ^2^*) was used to compare the composition according to the groups of professions of the respondents with the composition of the invited participants, to identify whether it was possible to conclude on any differences found between the groups of professions. Data from the NoMAD questionnaire was treated with descriptive statistics. Answers from the NoMAD questionnaire were scored 1 (disagree) to 5 (agree), except for question 10, which was scored opposite because of the negative wording in the question [24,25]. A mean score was calculated for each of the professional groups for each question by taking the sum of all the answers and dividing by the number of responses. The higher the score, the better is the implementation perceived in that core construct. The NoMAD questions were divided into the four NPT core mechanisms and were then analyzed by examining descriptive statistics for each of the core mechanisms. Mechanism scores for each participant were created by taking their average score in each mechanism and dividing by the number of valid responses, which stopped data from being skewed where there were questions in a category that the respondent had not answered (missing responses). Higher scores represent better perceived implementation in relation to each mechanism [24].

The Kruskal–Wallis test was used to analyze differences in perception regarding the implementation among the groups of professionals in relation to each question in the NoMAD questionnaire. Finally, when all the mean scores—for the total of answers—for each sub-core constructs were made, findings from the qualitative study were compared with the quantitative results to see if there was a relation.

For comparison of two subgroups’ answers to the NoMAD questionnaire, a Mann–Whitney U test was used for the data that was not normally distributed, and a *t*-test was used for data with a normal distribution.

## 3. Results

Observations at the department of pathology in Odense gave a lot of insight into the changes and new working stations as a consequence of the implementation of DIPA. A laboratory technician gave a tour in the laboratory and pointed out all the changes and showed the new workstations. During the visit at Sectra Denmark A/S, the researcher was able to try the software in a test-version, and discover the conditions that the pathologists would work in. Based on these observations, a description of differences in the workflow before and after the implementation of DIPA was made, as shown in Table 2. The table focuses on the changes for the laboratory technicians and the pathologists. The laboratory technicians have been appointed to do the scanning. Therefore, the role of the secretaries regarding DIPA is modest and predominately covers tasks related to external revisions as well as uploading of material for review.

In all, 18 interviews were conducted with 2 secretaries, 4 leaders, including the project leader, 6 laboratory technicians, including a superuser, and 6 pathologists, including 2 superusers. The general picture from the interviews was that the management and the employees were positive towards the implementation of DIPA. An employee expresses this as follows: “*I’m really looking forward to it [DIPA]… so I think it’s totally great and (…) it’s the right direction, and (…) it’s a really good step, yes”*. Some emphasized the topic of storing the amount of data that DIPA entails, as follows: “*… having a technology that can (…) perform or handle the amount of data that we need now, considering it [the technology] isn’t very mature, makes it quite a pioneer project in many ways”.*

Some of the employees were a little skeptical about the time of implementation based on considerations of whether the technology was mature, as in the following statement: *“I think it’s the future, and (…) what we’re moving towards, so I’m not really worried. I just think it might be a little premature (…) I’m not sure that (…) the technology is quite there yet”*.

The questionnaire survey was distributed to 181 staff and was completed by 123 respondents (68%). Eight respondents (4%) gave some answers.

Most respondents were female (80.15%), and participation was predominantly by doctors (51.91%), with a majority of them being pathology specialists (69.12%). Of all the pathologists, almost two thirds (65.22%) had been working as a pathologist for over five years. The average seniority for the other employees was 13.42 years. For characteristics of the respondents, see Table 3.

The respondents were not significantly different from the persons in the whole sample population with regard to type of employee (χ^2^ = 5.34, *p* = 0.0692).

### 3.1. NoMAD Survey

Overall, the analysis of NPT and the four core constructs for the four departments in total showed very positive feedback. All scores were over 3 except for two sub-constructs. Less positive feedback is seen in collective action, contextual integration (total score 2.73) where staff were asked whether there are sufficient resources available to support DIPA, and in reflexive monitoring, systematization (total 2.72) where staff were asked if they are aware of reports about the effects of DIPA (Figure 1 and Table A2). Examples of these views were also found in the interviews, as follow: *“So we haven’t received any introduction at all, no. (…) and we haven’t received the final system yet. (…) so nobody has really seen the final DIPA system yet (…) therefore, no one really knows how the system works”* and another interviewee said *“… so there are not many extra resources to give, so it will be something like that people have a job function and have to go a little back and forth to make it come together in the training situations”*. The question about awareness of reports about effects of DIPA showed slight concerns, as follows: “*… to take the sting out of the worries, I would have liked maybe to get a bit of information on what challenges can occur… and what the challenges have been so far. I went to a meeting where they talked about the experiences with DIPA [from another hospital] where they spoke about it [challenges]. It was interesting to hear but it was like nobody wanted to talk so much about the challenges”.*

Descriptive analysis of the mean scores $\left(\bar{x}\right)$ of the four NPT core mechanisms for all departments and groups of professions in total led to following results: coherence ($\bar{x}$ = 82.94%), cognitive participation ($\bar{x}$ = 91.59%), collective action ($\bar{x}$ = 71.82%), and reflexive monitoring ($\bar{x}\;$= 71.48%). This suggests that there is no core mechanism that leads to unfavorable expectations and readiness among the staff prior to the implementation of DIPA. Based on further analysis of the 16 sub-constructs, areas of improvement were found for the different groups of professions (Figure 1) concerning skill set workability, where the secretaries were found to not agree much with the statements (this is not found to be significant). In three of the sub-constructs, significant difference was found among the professions regarding the perception of the implementation process.

Coherence, individual specification: “I understand how DIPA affects the nature of my own work” (χ^2^ with ties = 6.052, df = 2, *p* = 0.0485). The laboratory technicians agreed more with this statement compared to pathologists and secretaries.Cognitive participation, initiation: “There are key people who drive DIPA forward and get others involved” (χ^2^ with ties = 10.635, df = 2, *p* = 0.0049). Again, the laboratory technicians agreed more with the statement compared to the two other groups of professions.Reflexive monitoring, reconfiguration: “I can modify how I work with DIPA” (χ^2^ with ties = 9.972, df = 2, *p* = 0.0068). The pathologists agreed most with this statement.

Looking at Figure 1, four other sub-constructs are found to involve larger discrepancy between the three groups’ attitudes towards the statements. This is seen in coherence and the sub-construct communal specification (“Staff in this organization have a shared understanding of the purpose of DIPA”), collective action in the two sub-constructs interactional workability (“I can easily integrate DIPA into my existing work”), in skill set workability (“Work is assigned to those with skills appropriate to DIPA”), and finally in reflexive monitoring in the sub-construct individual appraisal (“I value the effects that DIPA has had on my work”). However, none of these discrepancies is found to be significant, as follows: (χ^2^ with ties = 4.871, df = 2, p = 0.088), (χ^2^ with ties = 0.945, df = 2, p = 0.6234), (χ^2^ with ties = 3.732, df = 2, p = 0.1548), and (χ^2^ with ties = 1.690, df = 2, p = 0.4295), respectively. In each of the mentioned questions, the secretaries disagree the most with the statements, except with the statement “I value the effects that DIPA will have on my work”, where they agree the most.

Comparison of answers to the NOMAD questions has been made between subgroups. Comparison of the answers from trained vs. non-trained laboratory technician, and comparison of answers from certified pathologists/consultants vs. residents has been carried out, but we have found only minor differences in both positive and negative direction between the groups. This could be a result of the small sample size in the subgroups and should be studied further in larger studies.

### 3.2. Supplementary Questions from the Survey

We found that 52.03% of all the staff answered that they had received training in DIPA. Furthermore, 75% of these were pathologists, 21.88% were laboratory technicians, and 3.13% were secretaries.

When asked about whether DIPA would affect their ergonomics, 64.23% answered yes, and 51.9% of these thought it would be in a positive way. None of the secretaries thought it would affect their ergonomics. On the other hand, 63.15% of the laboratory technicians thought it would affect their ergonomics negatively, whereas 56.66% of the doctors thought DIPA would contribute positively to their ergonomics.

When presented with the statement “There are benefits with DIPA” the staff answered as follows: 0% disagree, 0% partially disagree, 7% neutral, 24% partially agree, and 68% agree.

When presented with the statement “There are disadvantages of the use of DIPA”, the staff answered as follows: 2% disagree, 16% partially disagree, 17% neutral, 28% partially agree, and 37% agree.

The pathologists’ view on the functionality and possibilities within DIPA are shown in Figure 2.

Descriptive statistics on the correlation between age and worries about transforming from analogue to digital pathology showed no trend (Figure 3).

## 4. Discussion

In this study of expectations for and perceptions of DIPA prior to implementation, we found that the staff at the four departments of pathology in the Region of Southern Denmark had overall positive expectations and felt ready for implementation and the concomitant changes of the workflow. This was especially clear from the analysis of the four core constructs of the NPT. None of these constructs had a mean score under 70% of the maximum score. Furthermore, the total mean score for each sub-construct was found to be high, over 3, except for in two cases.

Compared to other studies also using NoMAD to identify factors relating to implementation that either inhibit or promote the daily use of a certain device in the clinic, the employees in our study had high and positive expectations, and, furthermore, seemed ready to implement DIPA. When examining the implementation of acute kidney injury e-alerts in hospitals in England, Scott et al. (2019) found mean scores for the four NPT mechanisms to be coherence ($\bar{x}=72.30\%)$, cognitive participation ($\bar{x}=76.40\%)$, collective action ($\bar{x}=66.50\%)$, and reflexive monitoring ($\bar{x}=68.80\%)$. When comparing our total mean scores for each NoMAD question to their findings, Scott et al.’s population was found to a have a higher score in 6 out of 20 questions—4 in collective action and 2 in reflexive monitoring [24]. Cook et al. (2020) used NoMAD to investigate implementation behavior in dental students in relation to a novel oral health risk assessment tool. They calculated mean scores for each NoMAD question for 3rd, 4th, and 5th year students. These results compared to our findings show that in only 2 out of 20 questions did the dental students score higher than the respondents in our study. In collective action, the 4th and the 5th year students scored higher than the employees at the departments of pathology. In reflexive monitoring, one question scored higher for all three grades compared to our population [28]. This might be due to the fact that the decision to implement DIPA was made top-down, and because the respondents knew that all the departments of pathology in Denmark will have to implement DIPA in the coming years. These facts might have made the staff think that there would be no alternatives. This may explain the positive expectations. Furthermore, some of the departments of pathology have worked with WSI in relation to archiving, and others to diagnose frozen sections over distances some years prior to the implementation of DIPA. Finally, as seen in Table 1, users have been involved in the preparatory phase of the implementation process, and it is shown that in complex adaptive systems, such as healthcare, employees tend to accept new changes based on their own logic rather than the views of others. This entails that they are more likely to accept change when they are involved in the process than when change is imposed upon them [26].

Therefore, user involvement might have contributed to a greater sense of ownership among the employees and more positive readiness. It must be noted that comparing results from different countries, concerning different devices and different types of healthcare professions, is difficult. More research is required to be able to conclude what is a low, normal, and high level with regards to employees’ views on a new technology and normalization of the technology using NoMAD.

A limitation of our study is that for some groups of staff, the sample size is too small. In several of the sub-constructs of NPT, the secretaries were found to agree less than the other groups of professions. However, the results were not significant. This could be due to the small number of secretaries compared to the number of pathologists and laboratory technicians. Moreover, the secretaries are thought to play a smaller part in the use of DIPA and, therefore, per se, be less involved in the forthcoming change than the other groups of professions. Furthermore, due to the departments’ different times of commencement, it was not possible to reach all the employees at the same point in time of the implementation process. Some received the questionnaire after commencement but, since it was very close to commencement, we presume it has not affected the results in any great way.

We reached a very high response rate of 68% completed and 4% partially completed surveys. This might be because the implementation of DIPA is regarded by the staff as an important topic. Moreover, since the implementation will affect their daily work life, they expectedly have a strong opinion about how DIPA might (or might not) impact the workflow and their motivation for performing their job. We found that most pathologists have responded to the survey (89.74% of all included pathologists). Several factors can explain this—it could be because the scientist that sent out the survey is a doctor, or because the pathologists have access to a computer almost the entire working day, whereas the laboratory technicians have a lot of different tasks that are not always carried out near a computer. However, the secretaries’ working station is also a computer, and only about half of the secretaries responded to the survey (54.55% of all included secretaries). Indeed, more laboratory technicians answered (60.64%) than secretaries. The pathologists and laboratory technicians are the groups of professions where this digital transformation might have the biggest impact, which could explain the higher respond rate from them.

All three groups of professions score high in the following statement from NoMAD: “There are key people who drive DIPA forward and get others involved”. This could indicate a positive perception of the use of “implementation ambassadors” as a central part of the implementation process [29,30,31]. The laboratory technicians agree most with the statement. This could be due to the way the departments of pathology are organized, and how visible the ambassadors from the different groups of profession have been.

A general agreement was identified among the whole staff that insufficient resources were allocated to support the implementation of DIPA. One reason for this viewpoint could be the fact that during the planned implementation process expectedly entailing a huge change, the departments would still have to keep up the daily workflow. Technological implementation processes may require the allocation of extra resources [32], such as time, employees, or money. Both inner and outer facilitators and barriers might have affected our findings, but, to identify such variables, more data collection is needed after the implementation has begun.

Examples of statements from the interviews are used in this article to illustrate the results from the NoMAD questionnaire. However, further analysis of the interview data is needed. Such an analysis will be made at a later stage.

Conducting both interviews and observations gave a great understanding of the impact of DIPA on the workflow and tasks of each professional groups. The NoMAD is a questionnaire with closed response categories. Therefore, the pre-interviews made it possible to get a deeper understanding of how DIPA would affect the employees and also where there was a need for additional questions related to DIPA. The NoMAD is translated into Danish, and we used this translated version for our study, but it can be questioned whether the meaning of the survey questions in English is preserved in the Danish translation. Another challenge related to NoMAD is that there are no instructions for how to analyze the results, and different studies analyses the data in different ways [24,25]. This makes it difficult to compare results across different implementation studies. Overall, however, the NoMAD tool offers an easily applicable model that enables the assessment of implementation processes at both an individual and collective level, and identifies inhibitors and promotors of the process.

Our study has shown that the employees at the departments of pathology in the Region of Southern Denmark have high expectations and are ready for the implementation of DIPA. The region and the heads of departments seem to face an implementation process without huge barriers. The employees seem to acknowledge the need for implementing DIPA, which is possibly a result of including and informing them in the preparation phase of the implementation. Further studies of the impact of the implementation of DIPA and the perception of the staff of the use of DIPA in the workflow will be made at two different times after the commencement of the implementation. This additional study will show whether the positive expectation of the staff in the departments of pathology was realistic or not.

## 5. Conclusions

By the use of the NoMAD tool, this study prior to implementation of DIPA found that the respondents overall had high expectations, as well as motivation and readiness for the upcoming changes. At two points, the employees were skeptical, concerning the following factors: allocation of resources and awareness about the effects of DIPA. Based on the findings, it seems to be very rewarding to include employees in the planning process, to recruit engaged ambassadors, and to make sure to inform and include all groups of professions. It is important to make sure that the employees know not only how the intervention will work, but also why the intervention is necessary to implement and how it will be valuable in daily work routines.

Gaining insights into how staff at the departments of pathology experience and perceive these changes prior to implementation has provided valuable information for successfully leading and carrying out digital transformation processes in general.

This study has illuminated employees’ expectations and thoughts prior to the implementation of DIPA. The findings provide both relevant and important insights for future implementations of DIPA. It is our hope that other departments, regions, and countries will look at these results and utilize them in their planning of implementation of DIPA. In particular, they should be used managerially to localize if employees feel insecure, and to determine what to focus on to ensure a good implementation of DIPA.

## Figures and Tables

**Figure 1 ijerph-19-07253-f001:**
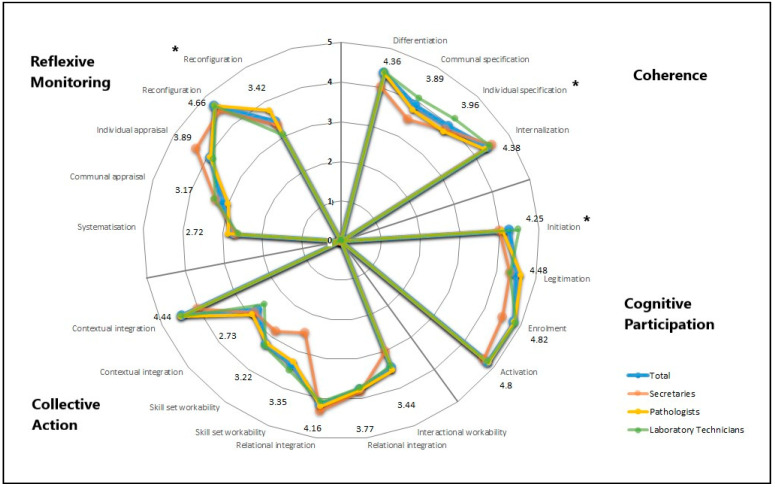
Petal chart showing the mean scores for each of the 16 NPT sub-constructs for the staff in total, and for each of the 3 groups of professions. Sub-constructs marked with an asterisk (*) indicate that there is significant difference in perception among the three groups of profession. The scores written are the number for the total scores.

**Figure 2 ijerph-19-07253-f002:**
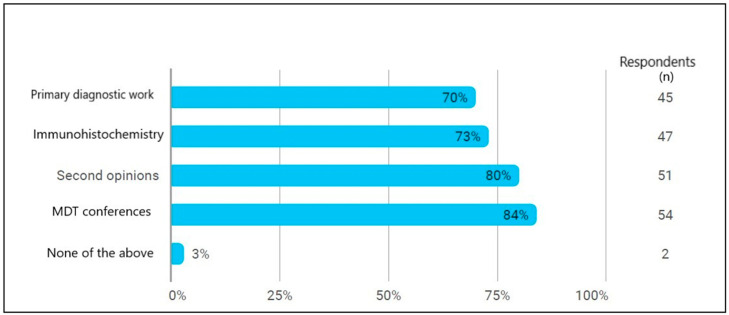
The pathologists’ responses to the question: “How likely is it that you personally would use digital pathology for the following clinical uses?” It was possible for each respondent to answer yes to several of the options. In all, 64 pathologists have answered the question.

**Figure 3 ijerph-19-07253-f003:**
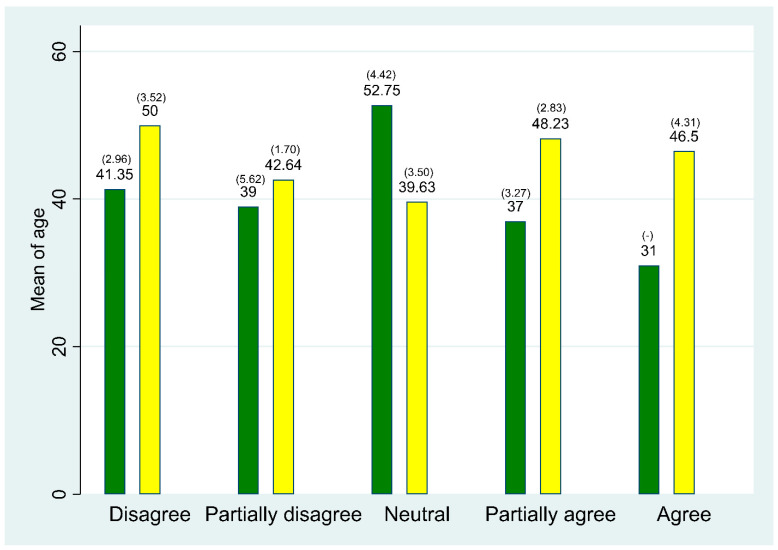
The distribution of mean age in each professional groups when answering the following question: “It worries me having to transition from traditional pathology to DIPA”. (SE). Green: Laboratory technicians. Yellow: Pathologists. Due to a too small group of respondents, the secretaries are left out of the figure.

**Table 1 ijerph-19-07253-t001:** A brief step by step overview of the whole implementation process, from the first acceptance of the idea of implementing DIPA in the Region of Southern Denmark to commencement.

Period		Main Subject	Groups/Committees	Participants
Late 2017	Preanalytic phase	Discussion of possibilities and feasibility	Project group Steering committee	Department heads IT Selected staff from departments
2018		Preparation of requirements specifications for scanners and for image management system (IMS)	Project group Steering committee IMS group Scanner group Groups for digital working space, and coordination between histology labs	Pathologists and laboratory technicians
2018		Bidding rounds for IMS	Project group IT	Pathologists and laboratory technicians IT
2019		Bidding rounds for Scanners	Department of medical technology	Medical technologists Pathologists and laboratory technicians IT
2019		Negotiations with companies	Negotiation group	Selected people from project group, medical technology, IT and steering committee
2019		Adjustment of scanner requirements	Department of medical technology Pathology departments	Selected people from these two groups
2019		Second bidding round for scanners	Department of medical technology	Medical technologists Pathologists and laboratory technicians IT
2019		Contract for IMS	Project group Steering committee	
2020		Contract for scanners	Project group Steering committee	
2019–2020		Preparation for IMS	Project group IT Pathology departments	
2020		Local preparation for digital pathology: technical, ergonomical, testing, training	Pathology departments	Laboratory technicians Pathologists Secretaries
2020		Preparation for scanners	Project group IT Medical technology Pathology departments	
Nov 2020		Upstart of digital pathology	All	

**Table 2 ijerph-19-07253-t002:** The working processes from tissue to diagnosis in a department of pathology before and after the implementation of DIPA.

Glass Slide Histology Pathways	
**Working flow before and after the implementation of DIPA**	
*Before*	*After*
** *The laboratory* **	
*The reception*	
The tissue or organ is received in a container at the department of pathology. The container is equipped with a barcode for further identification.	No change
*The grossing*	
The organ or tissue is examined by a pathologist or a laboratory technician and areas of interest and areas crucial for further diagnostic are cut out and placed in cassettes with the identification barcode. A macro description is made, and it is described what is in each cassette.	Similar to the workflow before DIPA, but now there is a further special focus on the amount and size of tissue in the cassette. There must not be too much tissue—especially not in the width.
*The preparation of tissue*	
The cassettes with tissue will be dehydrated in different concentrations of ethanol.	No change
*Embedding*	
The dehydrated tissue will be embedded in paraffin. Automated embedding will occur as usual prior to DIPA.	Similar to the workflow before DIPA, but now manual embedding will require that only certain cassettes can be used. The large format cassettes should be avoided.
*Sectioning*	
The paraffin embedded tissue is cut into very thin slices and placed on an objective glass. The automated cutting on the robot will be as prior to DIPA.	The workflow will be as prior to DIPA, but now in manual sectioning, one must be especially attentive to how the tissue is placed on the glass. The tissue must be placed right in the middle, as further lying tissue is at risk of not being scanned due to the scanner profile and its limitations. When one is handling megasections, it is also important that the tissue is not placed in the top of the glass, as the tissue in that situation not will be scanned/understood as tissue but as a barcode. The laboratory will have to standardize the placing of the tissue, as it has to be defined in the software where the barcode is placed and where the scanner will find the tissue.
*Staining*	
When the tissue is placed on the objective glass it will be stained according to what the pathologist has ordered. If it is special staining or immunohistochemistry, then the stainings will be quality assured on a microscope before the glass can continue the process.	No change
*Oven*	
**-**	New workstation. When the glass slides are stained, they will be placed in a heating cabinet at 60 degrees for 15 min to avoid the risk that the glass slides will be stuck in the scanner rack, as the coating material sticks to the rack if it is still wet.
*Scanning*	
**-**	New workstation. All glass slides will now be scanned. The scanners for megasections can scan 30 glass slides at a time or 60 “normal” slides at a time. The ordinary scanners can take up to 360 slides at a time.
*Dicomisering*	
**-**	New workstation. After the glass slides have been scaned the files or whole slide imaging (WSI) will be sent to a server and converted to the DICOM file format.
*Quality assurance 1*	
**-**	New workstation. The scanned and converted material (WSI) will be quality controlled. Here the focus is on the scanning, whether the picture of the tissue is clear, if it is possible to see the cells, or if the scanner has focused on dirt or grease on the slide instead of the tissue.
*Quality assurance 2*	
It is controlled that the tissue agrees with the macro description (measure, right type of tissue, the cutting—not too thick or folds in the tissue).	The principal is still the same as prior to DIPA, but now the quality assurance is made on the WSI. The laboratory technicians working with staining will have to quality assure the staining in the software program and not on a microscope.
*Distribution*	
The tissue sections will be distributed to the pathologists. The glass is put on trays and put on the respective pathologist’s shelf.	The WSI will be distributed on a computer to the pathologist’s pathology program. In the beginning, the pathologist will receive both the physical slides and the WSI on the computer.
** *The pathologists* **	
*Receiving the case*	
The pathologist can see in a software program that a case is ready. The pathologist has to go and collect the case (slides on a tray) from a specially assigned shelf.	The pathologist receives the case on the computer in a new software program. They click on the case and the WSI opens.
*Diagnostics*	
The pathologist will take each slide and look at it in the microscope.	The pathologist clicks on each WSI and can also scroll between each WSI.

**Table 3 ijerph-19-07253-t003:** Characteristics of respondents.

n (%)	Secretaries (n = 6)	Pathologists (n = 68)	Laboratory Technicians (n = 55)	Other * (n = 2)
**Sex**				
*Male*	0 (0)	16 (23.53)	8 (14.55)	0 (0)
*Female*	6 (100)	51 (75)	46 (83.64)	2 (100)
*Do not want to inform*	0 (0)	1 (1.47)	1 (1.82)	0 (0)
**Age**				
*Mean (lowest, highest obs)*	56.6 (48, 63)	45.42 (29, 66)	39.85 (25, 61)	63
*Do not want to inform*	1	18	15	1
**Department**				
*OUH*	3 (50)	40 (58.82)	32 (58.18)	1 (50)
*SHS*	0 (0)	7 (10.29)	10 (18.18)	0 (0)
*SVS*	3 (50)	8 (11.76)	11 (20.00)	1 (50)
*SLB*	0 (0)	13 (19.12)	2 (3.64)	0 (0)
**Seniority**				
*Mean (lowest obs, highest obs)*	20.1 (6.5, 38)		12.4 (1, 43)	22 (3, 41)
**Pathologist (specialized doctor)**				
*Yes*		47 (69.12)		
*No*		21 (30.88)		
**Years as pathologist**				
*0–5 years*		15 (31.91)		
*6–10 years*		8 (17.02)		
*11–15 years*		9 (19.15)		
*16–20 years*		6 (12.77)		
*21–25 years*		4 (8.51)		
*26–30 years*		3 (6.38)		
*Do not want to inform*		2 (4.26)		
**Management function**				
*Yes*	0 (0)	4 (5.88)	7 (12.73)	0 (0)
*Somewhat*	0 (0)	16 (23.53)	3 (5.45)	0 (0)
*No*	6 (100)	48 (70.59)	45 (81.82)	2 (100)
**Missing**				
	**Secretaries (n = 5)**	**Doctors (n = 12)**	**Laboratory technicians (n = 41)**	**Other (n = 0)**
*Some answers* *(Those who gave some answers are also included in the number of repondents above)*	0 (0)	4 (5)	4 (4)	0 (0)
*None reply*	5 (45)	8 (11)	37 (39)	0 (0)

* Laboratory technicians with functions such as teaching and specialized functions in the laboratory.

## Data Availability

Data from the study can be required by contacting the corresponding author M.L.N.M.

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
