# Peer review of "Prior to Implementation of Digital Pathology—Assessment of Expectations among Staff by Means of Normalization Process Theory"

_ijerph, 2022, doi:10.3390/ijerph19127253_

Round 1

Reviewer 1 Report

While the article is well written and presenting the value of implementing digital pathology, it seems to be longer than usual and some readers might get distracted away from the main point of the study.   

The study talks about the survey the group designed to implement digital pathology.

The topic is relevant to the field of digital pathology. However, the group emphasized on the survey. To make the article more informative I would suggest talking about the technical aspects on how they eventually decided on specific machines and softwares (such WSI scanner, softwares..).

The group is talking about their experience which might add value to others.

The conclusions are consistent with the evidence and arguments presented in the paper.

Some of the provided references are not in English but they might be helpful to the speakers of that specific language.

Reviewer 2 Report

This paper presents a study of staff reaction to the adaption of digital pathology (DIPA).  The topic shown in this paper is interesting, and its necessity is indisputable for the successful implementation of DIPA. This type of study will motivate other institutions towards DIPA which is essential especially when we have a shortage of pathologists worldwide and the number of pathology tests is increasing over time.

However, the reviewer strongly recommends a minor revision considering the following facts:

  1. The purpose of this study was to understand the expectation and readiness of employees. Therefore, it is crucial to clarify the understanding level of the employees about the changes in their roles in DIPA. The reviewer demands a more detailed explanation of how the technician, pathologists, residents, and other staff were introduced to their roles in the DIPA workflow. This explanation will help the readers to compile the findings of this study as well.
  2. A more intuitive introduction of the DIPA in contrast to the traditional pathology will be helpful for the readers.
  3. The introduction should also explain the importance of this study for the implementation of DIPA.
  4. The author should also mention how the findings of this study will be utilized for the implementation of DIPA.
  5. Why the author didn’t consider sentiment analysis to interpret the responses of the interviewees

Reviewer 3 Report

In this manuscript, the authors aim to assess the expectations and readiness among employees and management toward implementing Digital Pathology (DIPA), including their thoughts and motivations for starting to use DIPA. The authors concluded that it is very rewarding to include employees in the planning process, recruit engaged ambassadors, and inform and include all groups of profession.

The manuscript is of sufficient importance; however, some issues need to be addressed below.

Remarks to the authors

11-      When was the DIPA kick-off? Was it Nov. 2020 as per Table 1? In lines 22-25, the author mentioned that the questionnaires were e-mailed to 181 employees three months before DIPA kick-off, while the semi-structured interview used to design the questionnaire took place from Nov. 2020-Jan. 2021 as in lines 185-187. Please clarify. Further, please comment on which groups had a short interview (e.g 33 min) and a lengthy interview (101 min).

22-      Like the previous point, please clarify when the NoMAD survey was conducted, before or after the DIPA kick-off?

33-      Table 1 shows that teaching of laboratory technicians, Pathologists, and secretaries took place in 2020, Did teaching here refer to training?

44-      The current study involved four groups, as per Table 3, with Pathologists and laboratory technicians representing the majority of the study population. Despite being only 52.03 % of all the staff had received training in DIPA (lines 313-314), it does not seem to impact positivity and high expectations significantly. Have the authors explored why this important factor (training) does not have a significant contribution?

55-      While Table 2 showed that most of the upcoming changes are technical, i.e. require trained laboratory technicians, trained staff only showed 21.88% laboratory technicians compared to 75% pathologists, lines 313-314. Have the authors tried to pull out and focus on the laboratory technician group comparing the results of trained vs. non-trained?

66-      Have the authors tried to pull out and focus on the pathologists comparing certified pathologists/consultants vs. residents for any significant differences?

77-      Indeed, some significant factors might shape the response to the questionnaire and in turn, readiness and positivity to implementing the new technology. Have the authors considered the following:

A.    Familiarity with the technology to different groups? Pathologists are expected to be familiar with DIPA since the concept of digital pathology goes back to the 1960s, and as former medical students, they are familiar with the concept of digital imaging either through textbooks or online resources. Remote access to the images confers a more flexible work environment? Were there any observed differences between certified pathologists/consultants vs. residents, given the different workloads?

B.    There is a well-known Occupational Hazard for Pathologists in the form of Musculoskeletal Disorders due to prolonged microscope use. Would DIPA provide a more convenient, hazardous, less opportunity that enhances positivity and expectations?

88-      Please clarify how the workflow for secretaries was changed/modified by the implementation of DIPA. If not, please describe the relevance of this group to the current study?

99-      Lines 233-245 are better represented in a table format. Which group was a little skeptical?

110-   According to the study, the majority have high and positive expectations for the new system, yet some were a little skeptical. Since the use of DIPA alters steps in the workflow, some of the human tasks will be replaced by automatic digital transmission in the distribution of glass slides from the laboratory to the pathologists (lines 116-118). Did the author factor in any potential employees’ fear of losing their jobs in the future with the implementation of advanced technology, particularly those who were a little skeptical?

111-   In lines 74-76, the authors mentioned, “All four departments are involved and included in the digitalization of pathology in the region. The total number of certified pathologists is around 55 and about 20 residents”, a total of 75. While in Table 3, the numbers are 68 respondents, and 12 did not respond, a total of 80. Please clarify.  

112-   Lines 248-249, Most respondents were female (80.15 %), and participation was predominantly by doctors (51.91 %) with a majority of pathology specialists (69.12 %). The study took place in Pathology Departments; what other specialties are there? Please clarify.

113-   The manuscript will benefit from another round of editing. Some data are better presented in tables.
